# Exploring the Public Health and Social Implications of Future Curative Hepatitis B Interventions

**DOI:** 10.3390/v14112542

**Published:** 2022-11-17

**Authors:** Jack Wallace, Jacqueline Richmond, Jessica Howell, Behzad Hajarizadeh, Jennifer Power, Carla Treloar, Peter A. Revill, Benjamin Cowie, Su Wang, Mark Stoové, Alisa Pedrana, Margaret Hellard

**Affiliations:** 1Burnet Institute, Melbourne, VIC 3004, Australia; 2Australian Research Centre in Sex, Health and Society, Latrobe University, Bundoora, VIC 3083, Australia; 3Centre for Social Research in Health, UNSW, Sydney, NSW 2052, Australia; 4Department of Epidemiology and Preventive Medicine, Monash University, Melbourne, VIC 3004, Australia; 5Department of Medicine, University of Melbourne, Parkville, VIC 3010, Australia; 6The Kirby Institute, UNSW, Sydney, NSW 3052, Australia; 7Victorian Infectious Diseases Reference Laboratory, Royal Melbourne Hospital at the Peter Doherty Institute for Infection and Immunity, Parkville, VIC 3010, Australia; 8Department of Microbiology and Immunology, University of Melbourne, Parkville, VIC 3010, Australia; 9WHO Collaborating Centre for Viral Hepatitis, Peter Doherty Institute for Infection and Immunity, Parkville, VIC 3010, Australia; 10Center for Asian Health, Saint Barnabas Medical Center, RWJBarnabas-Rutgers Medical Group, Florham Park, NJ 07039, USA; 11Department of Infectious Diseases, Alfred Health & Monash University, Melbourne, VIC 3004, Australia

**Keywords:** hepatitis B virus, hepatitis B cure, implementation, public health, elimination, treatment, social impact, stigma

## Abstract

Hepatitis B is a significant global health issue where the 296 million people estimated to live with the infection risk liver disease or cancer without clinical intervention. The World Health Organization has committed to eliminating viral hepatitis as a public health threat by 2030, with future curative hepatitis B interventions potentially revolutionizing public health responses to hepatitis B, and being essential for viral hepatitis elimination. Understanding the social and public health implications of any cure is imperative for its successful implementation. This exploratory research, using semi-structured qualitative interviews with a broad range of professional stakeholders identifies the public health elements needed to ensure that a hepatitis B cure can be accessed by all people with hepatitis B. Issues highlighted by the experience of hepatitis C cure access include preparatory work to reorientate policy settings, develop resourcing options, and the appropriateness of health service delivery models. While the form and complexity of curative hepatitis B interventions are to be determined, addressing current disparities in cascade of care figures is imperative with implementation models needing to respond to the cultural contexts, social implications, and health needs of people with hepatitis B, with cure endpoints and discourse being contested.

## 1. Introduction

Hepatitis B is a global health issue with biomedical, economic, public health, and social impacts affecting individuals, communities, and health systems. The World Health Organization (WHO) estimates 296 million people were living with hepatitis B (3.5% of the global population) in 2019 [1], with annual deaths resulting from the infection estimated to range from 555,000 to 887,000 deaths [1,2]. Hepatitis B disproportionately affects the Asia Pacific region where 45% percent of the global population with hepatitis B live and where deaths from viral hepatitis outnumber those of HIV, tuberculosis, and malaria, combined [3,4].

One in five people with hepatitis B will develop cirrhosis or liver cancer without the appropriate guideline-based clinical monitoring recommended for all people with hepatitis B [5]. While vaccination effectively reduces hepatitis B transmission [6,7], and current lifelong treatments reduce liver damage and liver cancer incidence [8], a cure for hepatitis B is essential to achieve hepatitis B elimination goals [9,10,11]. Hepatitis B cure science aims to develop interventions that resolve hepatitis B infection and prevent progressive liver disease and liver cancer [11,12,13,14] with a functional cure described as the “sustained durable HbsAg loss with undetectable serum DNA allowing treatment cessation” and where a complete cure would eradicate “the virus, and all its replicative intermediates” [15]. The potential for curative hepatitis B interventions requires an examination of the social and public health implications of their implementation.

The WHO, and Australian and other national governments have identified elimination goals within public health strategic responses to viral hepatitis [16,17,18], however, there has been little progress toward achieving hepatitis B targets. While WHO elimination goals aim for 90% of people with chronic hepatitis B being diagnosed, and 80% of eligible people being treated, by 2030, most recent estimates are that 10.5% of people with hepatitis B were diagnosed in 2016, and only 2% of people with hepatitis B were being treated [19]. In Australia, while a far greater proportion of people are estimated to be diagnosed (73%), only 23% are engaged in care for the infection, and only 11% of all people with hepatitis B were being treated in 2020 [20]. 

The highly effective and well-tolerated curative treatments for hepatitis C have transformed global and national responses and the lives of people who access this cure [21,22,23,24,25], with critical social and ethical elements of a hepatitis C cure delivery occurring, in part, in response to hepatitis C cure implementation [26,27,28,29]. Hepatitis B has complex biomedical and social impacts including marginalization resulting from stigma and discrimination within many countries affecting elimination goals [9,30]. These social impacts are affected by incomplete or incorrect knowledge of the infection intersecting with cultural and linguistic diversity, traditional health beliefs, and a complex biomedical description. 

For people living in countries with an intermediate or high prevalence of hepatitis B, the infection presents itself as an intergenerational, chronic infection with economic, social, and sometimes fatal, consequences [31]. Most people with hepatitis B do not speak English and live within cultures without an equivalent word for hepatitis or virus [32,33,34]. This results in a disjuncture between hepatitis B as a “virus that invades bodies” [35], with understandings of ill-health only occurring alongside symptoms [36,37], or as a result of generational misfortune or breaching social norms [38,39,40,41,42], all of which affect trust in health care providers [43]. Traditional explanatory models of health play an important role for the communities most affected by hepatitis B with purported hepatitis B “cures” being marketed in many Asian and African countries [44,45,46,47,48,49] with traditional Chinese herbal remedies lowering liver disease/inflammation markers [50].

A focus on the biomedical and foundational literature for the hepatitis B cure has more recently expanded to examine the role of people with hepatitis B within cure science [51,52] including perspectives from people with hepatitis B about the elements and priority of the cure [51,53], the economics of cure access [10,54,55], and of regulatory responses supporting cure access [30]. Understanding the social and public health implications of curative hepatitis B interventions will help drive demand for any potential cure from community and healthcare providers. This research addresses a gap in the literature by identifying essential elements and possible challenges and enablers to implementing hepatitis B curative interventions.

## 2. Materials and Methods

This research was overseen by an advisory committee of hepatitis B experts with experience in hepatitis B cure science, specialist and primary health clinicians, hepatitis C public health implementation experts, social scientists, and community advocates including people with hepatitis B identified through the researchers’ professional networks.

Exploratory semi-structured qualitative interviews were held with key stakeholders identified by the advisory committee. The participants held responsibilities or contributed to global, regional, or local public health policy and program development, scientific and/or clinical investigation or translational research, or engagement with the populations most affected by hepatitis B. Of 43 emailed invitations, 31 key stakeholders agreed, with recorded verbal consent, to participate in an electronically recorded interview, nine of whom also participated in the advisory committee. Interview prompts, pilot-tested with advisory committee members included:What is your understanding/what do you know of hepatitis B cure science?What are your personal expectations of what a cure could provide?What are the social/political/economic/ethical issues related to the development and implementation of a hepatitis B cure?From your perspective, are there structural barriers to implementing the cure at a global, regional, national, or local level?How do we achieve equitable access to the cure? What do we need to do to achieve equitable access to the cure?Are there health systems that need to be strengthened to enable a rollout of a cure? Who should be the first to receive the cure?What will the role be for the vaccination program in an era where hepatitis B can be cured?What would the impact of a hepatitis B cure be on individuals and families?

Interview recordings were professionally transcribed verbatim, checked for accuracy, and de-identified. The analysis for this article was conducted by the first author (JW) after multiple individual readings of transcripts, and, given the exploratory nature of the interviews, inductive thematic analysis was used with the stages as described by Braun and Clarke [56,57], and data coded using NVivo 12 (QSR International Pty Ltd., Hawthorn East, VIC, Australia). Field notes were used to support the development, organization, and prioritization of the themes in relation to their relevance to framing issues related to the public health implementation of a hepatitis B cure. Ethical approval for the study was provided by the Alfred Hospital Ethics Committee (178/2).

## 3. Results

Thirty-one people participated in this study with professional responsibilities and expertise at global, regional, and national levels, and employed within research, government, non-government, and pharmaceutical institutions, and who resided in the Asia and Pacific region (5), including Australia (14), and from Europe (3), Africa (2), and North America (7). Participant expertise included virological and/or molecular science research (described in the results as basic science); primary health care and specialist clinicians; social, epidemiological, and public health researchers; policy and community-based advocates. Participant expertise often extended from their primary professional role into various forms of advocacy and policy participation, with six participants having personal experience of living with hepatitis B. Given the ethical need to ensure confidentiality for participants, a general professional descriptor is used to identify quotes throughout this manuscript. The key themes identified and informed by the WHO health system building blocks are:Describing cure scienceHealth system preparationRequired clinical infrastructureEquity and AccessBeyond the “patient”: the social and individual implications of the cure.

### 3.1. Cure Science Narratives

The endpoints used by participants to explain hepatitis B cure fall under two previously well-established descriptions: a “complete” cure and a “functional” cure. Among key stakeholders, the descriptions and assumptions made about cure reflected their professional expertise and frameworks, and the direct level of engagement with, and confidence in their knowledge of hepatitis B cure science. A broad range of other cure descriptions is used within literature including sterilizing, complete, partial, virological, durable, absolute, immune cure, true, remission, and finite. One basic scientist unambiguously described a cure as a *finite treatment*, while a clinical scientist was clear in their expectations of a complete cure that they determined would not occur *in our lifetime and possibly in anybody’s lifetime*.

The impact of the cure for other participants specifically related to their professional intersection with hepatitis B, with one primary care physician noting that the cure reduced the requirement to access clinical services, with *no future monitoring, and no future treatment*. Another clinical and public health researcher described the cure as meaning that people with hepatitis B, *don’t have to engage with the health system anymore on that issue*, while a social scientist framed the cure as *providing people freedom from worrying about the implications of hepatitis B in their life*. While one advocate with hepatitis B detailed their clinical-related expectations of the cure as not needing to *take any medications; you don’t need to get monitored*, another advocate noted the intersection of the cure with human rights and biomedicine, while tacitly reflecting a priority of these outcomes.


*I’m happy for, one, the clearance of surface antigen so that a person would not be identified in any system … and two, the assurance of not developing liver cancer.*


The discourse used to describe hepatitis B cure aims and outcomes are defined by basic scientists with little involvement of people with hepatitis B or their communities, with one advocate noting cure outcomes as *mostly defined by clinicians and virologists with little to no real consultation with the affected community*. Additional concerns related to the cure discourse were raised by a basic scientist in the phrase “functional”: *I don’t like the term functional cure, because if someone said to me “you’ve got a functional cure”, I’d say “Well what’s a non-functional cure?” It’s a meaningless term*.

The language used to describe the cure could affect implementation within many communities, with one public health expert reflecting on needing to understand the context in which people with hepatitis B currently respond to the physical impact of their infection, *I’d hate to see us jump and start using the word ‘cure’ prematurely and lose the credibility with the community*.

### 3.2. Health System Implementation

Overwhelmingly, participant narratives emphasized the need to prepare health systems for curative hepatitis B interventions, with systems currently focussing on hepatitis B prevention or clinical management of chronic infection reorientating to a primary goal of cure access. One public health expert suggested reframing cure from an issue of basic science into a broader public health and viral hepatitis elimination framework so that: *you could have that bigger picture of eliminating a disease; … cure should fit within that kind of …narrative*. Focussing on implementing the hepatitis B cure was seen as providing a paradigm shift in understanding and responding to hepatitis B, particularly in responding to lessons gained through hepatitis C cure implementation including proactive case finding and linkage to treatment, and as noted by a public health researcher, *it would be a huge help to hepatitis B … to be armed with a new tool and work through … all the learnings and mistakes from hepatitis C, and how can we do better in B*.

The few countries currently on track to reach global hepatitis C elimination targets are mostly high-income countries with robust public health systems [58]. One public health implementation expert noted that forward planning to scale up access to services and simplify clinical pathways could have fast-tracked progress: *What we are doing now in hep C is we’re going back in time, perhaps doing things that we should have done in preparation for the cure*. Forward planning was also supported by an advocate who reported that it was essential to *think around access from early on, like not going to access down the road*.

Discussion on cure implementation models highlighted deficits in current health system responses to chronic hepatitis B. One basic scientist noted that *if we developed a cure tomorrow, it would not make a dent in the cases*, while a clinical scientist anticipated that the availability of a cure would increase the willingness of people at risk to be tested for hepatitis B: *People would come up and say, “Okay, actually now that you’ve got a cure, I want to get tested and find out whether I’m hep B positive.”* One policy expert noted the need for proactive community development to drive demand and improve health service access to ensure that these communities and their healthcare providers are aware of, and able to access, the cure when it becomes available: *We’ve still got to get people out there to come in, get tested and get treated, and then you prepare them, if the cure comes. Then the cure comes, and they will be the first.*

Forward planning for hepatitis B cure implementation supported equity as well as broad access. As with most issues related to hepatitis B, complexity is at its core including an intersection between cure science, access, cost, equitable implementation, and ethnicity. One policy expert described hepatitis B as a *racially related epidemic: it’s not white*, with the implication for cure implementation needing culturally relevant and accessible models of care, which was highlighted by one advocate noting *you can’t do hep B cure for white Caucasians, it’s not going to work*.

All participants supported the continued development and strengthening of hepatitis B vaccination program implementation, both in its specific prevention roles and as being pivotal for viral hepatitis elimination: *We need a vaccine …for hepatitis B and hep C, to get anywhere near elimination (Public health expert)*. This was reinforced for one social scientist with the experience of HIV, where treatments transformed the infection from one with high mortality to a chronic disease, and where reducing access to prevention interventions had significant outcomes.


*The theory of elimination is …treatment plus prevention… we’ve seen examples … in terrible outbreaks of HIV in certain communities, because prevention has fallen off.*


One public health expert saw hepatitis B elimination, incorporating vaccination and curative hepatitis B interventions occurring within a continuum,


*Prevention is absolutely as critical as cure … we will never capture everybody. … Elimination is not a moment in time: it’s a continuum, it’s a spectrum, you just move constantly in between.*


### 3.3. Required Clinical Infrastructure

The clinical infrastructure required for hepatitis B cure implementation will depend on the complexity of the intervention including how it would be administered. Without knowing the form of this regimen, assumptions were made based on the experience with hepatitis C. One clinical scientist anticipated the infrastructure required for hepatitis B cure implementation would be more complex than for hepatitis C.


*You have to have an infrastructure system for finding people who have hepatitis B;… testing to … confirm the infection … monitor them on the treatment, … confirm whether they get a cure, and monitor them for a longer period of time to declare a cure.*


Hepatitis B cure implementation and planning will need to involve primary health and community care services, with one basic scientist noting *we need non-specialists managing people*. For several participants, nursing staff will be integral to delivering the cure, with one advocate and a public health expert noting the experience in developing decentralized responses for hepatitis C.


*The hep C cure was complicated when it started. … I’m not sure you will need specialists for this kind (hepatitis B) of management and treatment … for the most part, primary care providers, nurses, nurse practitioners, community doctors, barefoot doctors, that’s who should (manage the hepatitis B cure).*



*A tolerable cure … would leverage what we’ve learnt from hepatitis C, and a lot of the programs would be nurse led. It’s just oversight from the prescribing clinician.*


Using a broader range of health providers to implement the cure was highlighted by a public health expert and an advocate who noted the lack of specialists in Africa meant that cure implementation will inevitably require a broader range of health professionals, a point which has also been previously made in relation to Asian and Pacific countries [59,60,61],


*(Where) there’s no liver specialist in the entire country … who’s going to be managing people with hepatitis B? Even if you got everyone screened, what are you going to do with them?*


Two participants, including a clinical specialist, noted that cure implementation would be most successful if it drew on infrastructure already available within country health systems, particularly with synergies occurring given the nature of hepatitis B transmission from mother to child providing a site for cure implementation:


*For a country like Papua New Guinea, I’d probably work through … maternal-child healthcare clinics; … they’re probably the best functioning infrastructure in the country.*


While the prevalence of, and mortality from, hepatitis B vastly outnumbers that of HIV, particularly in the Asia Pacific and African regions, and where significant infrastructure had been developed to respond to HIV, one advocate was concerned that this infrastructure would not be available for cure implementation: *you have infrastructure, because they did it for HIV and yet we can’t piggyback on that, (there’s) an entire global public health infrastructure system that you can’t use for anything else*.

The focus on biomedical definitions and outcomes conflicts with the reality for most people with hepatitis B who access healthcare through traditional or cultural medical practitioners, as noted by several advocates: *A lot of people with hepatitis B rather patronize orthodox traditional healers than going to the hospital, that is the reality of what happens in [the] majority of African communities*.


*People would much rather spend their money on promised (traditional) cures than harsh, harsh Western medicines, which are more expensive, and that they don’t have access to… and then when there is a cure, who’s going to believe us?*


In reflecting on previous comments for the proactive preparation for cure, ensuring that the communities most affected were involved from the start of cure implementation was noted, with one social scientist stating the need for *a workforce that’s community connected, and community, culturally, competent … That’s not an afterthought or it’s not a phase two or three or four, it’s from start*. For one public health researcher, a lesson from hepatitis C elimination was the need to understand how the communities who most need the cure are *cognitively, emotionally, socially engaging in the notion of a cure….Just because you’ve got the hammer, everyone with hepatitis B looks like a nail…*

### 3.4. Equity and Access

Resourcing will be pivotal to access at individual, country, and global levels with a clinical specialist noting *the costs will determine what equitable access would look like*. Several participants described the need for the comprehensive global implementation of the cure particularly with hepatitis B primarily affecting countries without the capacity to make the health system adjustments necessary to provide this cure. For one public health expert, one lesson for hepatitis B cure implementation came from COVID-19, and was *we’re only as safe as our neighbourhood*.

The sheer number of people with hepatitis B and the audience for a cure vastly outweighs HIV and hepatitis C, with one policy expert recognizing that *the end stakes are much higher, because it’s three hundred million people now*. One clinical specialist reported the need for *a global solution, and not just a solution for the very small proportion in high income countries that have hepatitis B: not that they shouldn’t get the cure, but that everyone else gets it as well*. Another advocate highlighted that equitable access to any cure implied a role for global institutions, particularly in a context where hepatitis B primarily affects lower- and middle-income countries: *what’s the role of international organisations such as for example WHO, Global Fund, in making equitable access to the treatment possible*.

The lessons learned In the comprehensive and coordinated global response to HIV were highlighted by a public health expert, particularly where lower and middle-income countries experience the greatest burden of the infection: *Who’s going to pay for low-income countries to get cured? High income countries are footing the HIV bill largely; Who’s going to foot the HBV bill?* This perspective was supported by an advocate who noted that hepatitis B *was the biggest infectious disease crisis that low- and middle-income countries have been asked to tackle alone, without any … real support from any international institutions*.

Within several Asian and African countries, hepatitis B infection has significant social implications with one Asian-based advocate worried this marginalization could affect public support of funding curative interventions, *If the price for the treatment is … the same or maybe more expensive than the HCV treatment, I don’t think the community will accept (this)*. At a country level, a different perspective came from a clinical scientist and was highlighted by a United States-based advocate, who noted an essential impact of the lack of a national health service and where *there will never be equity, because (the health system is) private insurance based*.

Chronic hepatitis B is primarily transmitted from mother to child with most people with hepatitis B in industrialized countries having families living in multiple countries. This cross-national family scenario could be culturally and morally challenging for people with hepatitis B who would have access to the cure, while their families may be living in countries where curative interventions are unavailable. One lesson of HIV treatment access and hepatitis C cure implementation was of buyers’ clubs importing generic direct-acting antivirals with legal, economic, and quality implications. One public health implementation expert noted that *if people are provided with a cure here, as a rich nation, but not their families or contacts in their home countries …is there a black-market possibility for this stuff?*

Implementation of the hepatitis B cure can potentially be cost-effective using traditional cost-effectiveness models [10,54,55,62]. One public health expert noted the impact of hepatitis B on health security, particularly within lower- and middle-income countries, with implications for people coming from, often collectivist and migrant, communities living in industrialized countries.


*The individual economic benefit to not having me die, or grandpa die, or Aunty Sue die …of their hepatitis B, for some families is huge.*


Within conventional public health models, hepatitis B testing and diagnosis occur within healthcare services, which is challenged by the experience in many Asian and African countries where testing occurs within educational, workplace, and immigration visa settings. One implication of these testing processes is that, while diagnosis is cheap and accessible, and on the rare occasions where linkage or referral to health care services does occur, this access to clinical services includes a requirement to pay for testing and/or drugs, with one advocate noting.


*Everybody gets screened free because of the employers or because of the government program … but in between testing and treatment, you’ve got the diagnostics … to confirm if you need to … treat or not to treat, and most people get stuck in the middle. …you have to spend (money) for [monitoring].*


Hepatitis B viral genotypes affect natural history, disease progression, and treatment responses. An issue across all themes related to the impact of ethnicity, including in cure research and could impact equity. One basic scientist reported that the hepatitis B genotype D being primarily used for cure development—a genotype mainly affecting people in Europe, the Middle East, India, and Africa. The implications for cure development and implementation were that any cure may be of greater initial utility outside of the populations with a greater prevalence of the infection and provides another example of differential cure access across viral genotypes and populations.


*Genotype D is mostly Europeans and Americans. It’s again the rich countries that end up getting first go at it and a better go at it, but it may be that we have to concentrate on these populations before we can get enough additional investment into cure science to be able to extend that out to the rest of the world.*


One policy expert recognized the unconscious bias of gender in science, and questioned if hepatitis B was affected by unconscious bias in relation to ethnicity, particularly relating to hepatitis B research funding, *unconscious biases does happen, and we do not know how structural unconscious bias is when you actually put in the research grants. It will be there. We all know it, we all describe it*. This perspective was supported by one researcher who noted that *in Western Europe, we don’t really have hepatitis B*.

### 3.5. Beyond the “Patient”: Social and Individual Implications

As noted previously, cure descriptions reflected the professional frameworks and knowledge of the person talking about it. Within many Asia Pacific and African countries, hepatitis B has significant impacts over and beyond that of a medical condition. One Asian-based advocate living with hepatitis B highlighted the importance of ‘cure’ to address both the clinical and social impact of hepatitis B: *When the clinicians talk about it, it’s preventing you from getting liver cirrhosis, but the thing is the HBsAg would still be positive*.

A public health expert and a specialist clinician recognized the intergenerational impact of hepatitis B and described the impact of cure as not only affecting individuals, families, and broader communities: *It’d stop intergenerational … morbidity and mortality…. It would say now is the time that you don’t die like your parents died or like your grandparents died … it would be life changing at a population level as well as an individual level*.

Three participants recognized how the cure would particularly affect women and reduce the psychological impact and potential distress from feeling responsible for hepatitis B transmission. For one advocate with hepatitis B, curative interventions would reduce their mother’s guilt resulting from hepatitis B transmission.


*My mum strongly believes that she is the one who put me into this rabbit hole of getting the disease … If I can get the cure, I can … confidently … say, “Mum, take it easy, don’t worry about it. I will be treated.”*


Several participants, including one advocate, spoke of additional social and practical implications of the cure over and above that of any clinical impact and highlighted the real-world implications of cure goals and definitions. Within many contexts, this also highlights the critical role that screening plays in the lives of people with hepatitis B with its immediate and comprehensive social impact.


*Even if you’re cured from hep B, and if you still get screened for hep B, you still have that problem of stigma and discrimination, right? People would rather have that HBsAg turn into negative, which is more important.*


An alternative perspective to cure outcomes and definitions and their implications was provided with experience from a clinical specialist who reported that *in Africa … people want to be antibody negative, they want … no remnant of the virus… being antibody negative is really important to them*.

Most people with hepatitis B living in colonizing nations are born overseas, and where issues such as racism, housing access, employment, health access, and other settlement issues for those who are more newly arrived will continue even with the availability and access to any hepatitis B cure. One advocate noted that, while the cure would reduce self-stigma related to hepatitis B, other social issues would remain: *we’ll always have this sort of discrimination and those sorts of stigma, just not based on hep B*.

While the psychological impact of being cured of hepatitis B was recognized by many participants, additional issues related to the impact of hepatitis B on subjective understandings of identity, and how this could affect implementation, were noted by an advocate and a public health expert.


*Entire personalities have been built around having hep B in some communities. It worries so many people in so many ways that … it’s sort of inconceivable how it could change someone if a lot of their identity is tied behind it.*


One social scientist highlighted essential differences between the biological and subjective experience of hepatitis B, further complicated by the asymptomatic nature of the infection, particularly among populations whose health beliefs consider illness occurring only with symptoms.


*What it means to have an illness, particularly one that’s asymptomatic, is not going to be recognised through those clinical measures or clinical markers. … And the politics of listening to the voices of affected communities, and prioritising that with ethics and politics.*


These data identify that the implementation of potential hepatitis B curative interventions will only be successful with resourced and reorientated public health responses developed in partnership with at-risk communities, and centered on the needs and experiences of people with hepatitis B.

## 4. Discussion

A hepatitis B cure will be revolutionary, with impacts far beyond that of a virus being removed from an individual or in achieving strategic viral hepatitis elimination goals. While cure science steadily progresses, the true potential of this revolution will not be realized unless the cure is scalable and accessible for people with hepatitis B across the globe, especially those born in middle and low-income settings.

This study finds that public health systems and affected communities need pre-emptive and systematic preparation to implement curative hepatitis B therapies and identify and address current and anticipated barriers to health service access for people with hepatitis B and their communities. This would include developing new models of care for hepatitis B, and resolving health systems challenges to hepatitis B clinical management, including improving diagnosis systems and scaling up capacity, and re-engaging with people previously diagnosed and lost to care. Using a public health viral hepatitis elimination framework [10] to consider the variety of interventions, health system, social and community structures, and breadth of stakeholders required will support hepatitis B cure implementation planning and recognizes and validates that eliminating hepatitis B as a public health threat by 2030 will not occur without global access to curative interventions.

Hepatitis B is a global health challenge mostly affecting countries and communities with the least resources or capacity to respond to the needs of people affected [1,2,10,16]. Universal infant hepatitis B vaccination will remain essential to secure health gains made through curative interventions, with any cure supporting, extending, and reinforcing vaccination achievements. The vaccination program has made substantial achievements with its significant global, regional, and national resourcing; however, the same cannot be said of the systematic resourcing of responses to chronic hepatitis B.

The financial cost of curative interventions will dictate their availability and access, with equitable access requiring global or regional financing to support its implementation within lower- and middle-income countries [63,64]. At the same time, this is not to negate the challenges involved for people with hepatitis B accessing curative interventions in higher-income countries that do not provide access to universal health coverage, particularly given the social economic status of the communities most affected by hepatitis B in these countries.

The proportion of people with hepatitis B engaged within care is low [19,20], highlighting the need for comprehensive health promotion and community engagement activity along, and beyond, the cascade of care continuum. Increasing the number of people tested and diagnosed with hepatitis B, no matter where this testing occurs, and then being linked to clinical management services is critical, and this will directly affect the number of people with hepatitis B accessing curative interventions. Several participants supported developing community-based models of care for implementing curative interventions. An effective public health response to curative hepatitis B interventions will need to be established in partnership with affected communities and a broad range of health providers, and incorporate informed and responsive health promotion interventions and clinical service models [36,65,66]. The decentralization and simplification of care implicit in this preparation will need to be balanced with the level of expertise required to manage cure interventions, particularly in their initial implementation.

Ensuring full access to health services for people with hepatitis B requires developing their trust in health systems at individual levels and in the system as a whole [43]. It is difficult to persuade people with hepatitis B in many lower- and middle-income countries that clinical management of their infection is a priority while there are clear discrepancies in access between HIV and hepatitis B treatments. Determining country-level barriers, including systematically identifying and addressing issues that reduce health system access for people with hepatitis B, such as requiring payment for viral monitoring tests in contexts where screening is free of charge, and where treatment drugs are government-funded is imperative for equity.

The availability of curative interventions was suggested to increase the willingness of people with hepatitis B to access clinical services. This should not be taken for granted given the Australian experience, where hepatitis C cure availability, as the major strategy supporting access, proving to be inadequate, particularly among populations experiencing stigma, discrimination, and social marginalization. With slowing rates of testing and treatment in Australia, new strategies are needed to promote the hepatitis C cure among those yet to be treated to achieve elimination goals [67]. Participants, mostly those living with hepatitis B, noted the fear of social marginalization resulting from hepatitis B infection was a continuing barrier to implementing effective public health and clinical responses to hepatitis B.

It is anticipated that the availability of curative interventions will be staggered both in relation to eligibility, resources, clinical priorities, and efficacy. The implementation of curative interventions will not only benefit individuals, but fundamentally affect families where hepatitis B occurs across generations, and often across multiple countries. Curable interventions will not only interest individuals who access health services and to whom a cure is prescribed, but, as experienced with the hepatitis C cure and HIV treatments [68], the availability of the cure in one country will be of interest to citizens of other countries. It is difficult to predict the implications of this, but given the social and clinical implications of hepatitis B, it will be a challenge for individuals receiving curative interventions in one country while family members with hepatitis B in other countries are unable to get similar access. Given the global nature of hepatitis B infection, and the existence of cross-national communities, the staggered nature of implementation of cure access may have unanticipated outcomes that are worth investigating.

While biomedical and clinical perspectives on hepatitis B cure are well-established in the concepts and discourse used to describe them, these perspectives are less tangible when discussing cure and its impact among lay populations. The different descriptions of the cure from different personal perspectives and professional expertise highlight the different meanings and implications that hepatitis B cure can have. The implications of language are enormous for cure implementation. The biomedical description of hepatitis B is complex, and most people with hepatitis B—the recipients of the cure—are not bio-medically trained, nor will they necessarily understand the nuance of words such as “functional” when used to describe the cure, particularly when these terms need to be translated into languages other than English. In addition, the success of the hepatitis C cure and its simplicity will inform expectations of the hepatitis B curative interventions.

The term “cure” is heavy with assumptions from people without a specific biomedical framework for understanding the term. Cure science terminology has different meanings and implications for different audiences; there is a need to develop a shared language to ensure that a partnership approach where affected communities are central is developed to guide its implementation. Consistency in language is important, particularly when dealing with a condition affecting populations for whom English is not their first language, where the condition already has a complex biomedical description, and where scientific nuance is privileged.

Developing clear and inclusive cure descriptors by gaining a greater understanding of people with hepatitis B from a broad range of cultural and ethnic backgrounds will support its implementation. Redefining cure as a process or continuum rather than an outcome could lower expectations and enable improved engagement of people with hepatitis B with cure science.

Cure is a term that already exists in relation to hepatitis B, particularly in countries where traditional medicines provide accessible and culturally nuanced approaches. The promise of a biomedical “cure” needs to be cognizant that “cures” already exist for many populations, and these will affect the credibility and acceptability of any biomedical cure, particularly with questions of nomenclature, and unclear or unrealized expectations. Descriptions of a “functional” cure will have unintended consequences among people with hepatitis B, let alone health care workers given its previous use with HIV cure and its varied definitions. Further research is required to investigate the priority and nuances of the social impact of clearing surface antigen, supported by Asian-based participants, as opposed to an antibody-negative status supported by African-based participants.

Hepatitis B is an asymptomatic infection until late in the course of illness for those in whom advanced liver disease or liver cancer occurs. This is significant for many individuals and communities for whom the infection may not be as much of a priority as other health, social, cultural, family, economic, or settlement issues. For people with hepatitis B, the cure will be one part of their lives, and often a very small part of their lives although one that could have possibly significant implications later in life.

While a variety of backgrounds of participants were recruited to this study, a broader range of professional expertise and a larger sample would likely provide a similarly greater breadth of findings. One big gap in hepatitis B cure science is investigating how people with hepatitis B respond to the potential of cure. This study used semi-structured exploratory interviews, a process that is culturally relevant to the participants—professionals who have the confidence, language, concepts, and skills to talk about their professional experience and expectations in relation to cure. There is a chasm between the biomedical explanation of hepatitis B and the lived experience of hepatitis B as an asymptomatic, intergenerational, and sometimes fatal, liver condition, that because of its social impact requires a sophisticated understanding of disclosure. Investigating the perspectives of people about potential hepatitis B cures and cure science requires sophisticated co-design methods that are embedded in the broad and differing range of cultural experiences of people with hepatitis B, that will not cause harm, and that understands and responds to the complex intersectional nature of hepatitis B.

Eliminating hepatitis B as a public health threat by 2030 will not occur without global access to curative interventions. Further research to inform cure implementation is required with this study being an initial and exploratory step in foreshadowing some issues affecting this implementation in relation to hepatitis B.

## Data Availability

Data generated during the study are not publicly available for confidentiality reasons and may be available from the corresponding author on reasonable request.

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
