# Peer review of "Exploring the Public Health and Social Implications of Future Curative Hepatitis B Interventions"

_viruses, 2022, doi:10.3390/v14112542_

Round 1

Reviewer 1 Report

Overall assessment

·       This is an important study of key stakeholders’ perceptions of the social and public health implications of HBV cure in Australia. The paper is well-written, rich, thoughtful. The report proactively anticipates some key considerations for HBV cure research and implementation. The focus on cure science narratives, health systems implementation, clinical infrastructure, equity and access, as well as other social implications (including identities) is timely and a welcome addition to the literature.

Major comments

·       The paper title is not clear. It would be helpful to differentiate between ‘curative regimens’ and ‘treatment regimens’. Could the title be replaced with: Hepatitis B Cure Interventions: Exploring their Potential Public Health and Social Implications? Would also avoid the expression ‘curative hepatitis B treatments’ in the text since it is not clear if authors are discussing 'HBV cure’ or ‘HBV treatment’. This comment applies throughout the entire paper. Similarly, authors should be also be careful with the word ‘cure’ and make a clear distinction when they mean ‘HBV cure research’.

·       It may be helpful to include key definitions of ‘HBV cure’ in the Background section in addition to Results (as it would help better differentiate these with ‘treatments’).

·       Could the authors please make sure they have followed the Consolidated Criteria for Reporting Qualitative Research (COREQ) checklist for reporting of Materials and Methods and qualitative results?

Minor comments

·       Authors should carefully review the paper for typos and grammatical errors. For example, Page 1, Line 39: Should it be WHO estimates? Page 5, line 223: ‘strengthing’ vs. ‘strengthening’, etc.

·       Please check long sentence on Page 8, Lines 336 – 339: there appears to be a word or two missing.

·       Some of sentences are very long and could be shortened or streamlined. For example, see Page 8, Lines 343 – 345.

·       Page 9, line 407: Please check transition between text and the quote to make sure the text flows smoothly.

·       Please make sure to add punctuation where necessary. There were several places where punctuation was missing, particularly before quotes.

·       This is a matter of preference, but we would recommend ending the Results section with a brief transition instead of a quote.

·       In some places, it would be helpful for the authors to highlight key areas of convergence or divergence between stakeholders or stakeholder types.

·       Have the authors considered a framework or figure to summarize these important social and public health considerations, or a table summarizing the key points that emerged from the Results? This could help with disseminating Results with communities of interest, and advocating for greater attention to public health and social implications of HBV cure research.

Author Response

Thank you to the reviewer for these comments. Our responses to specific points are attached.

Reviewer 2 Report

Authors report on a study of interviews with HBV basic scientists, clinicians, persons with lived experience, public health and social epidemiologists. Their focus is on the public health and social implications of HBV cure, once it becomes available.

I think the topic is an important one to address and timely as we work out the details of how we even define cure and plan for what this may mean for planning around dissemination.

Overall, I thought the paper was well-written and an appropriate design to pull out some of those details and themes. I would have loved to see the issues more thoroughly outlined with perhaps more examples and comparators with how things were done for HCV and why HBV is different. That being said, the development of such a report may be of a larger scale than for a journal article and given the complexity of the issue perhaps can’t be distilled so readily.

Specific suggestions within the text:

Title – while it has a bold outline of “curative HBV treatments” (and I truly look forward to when we can use that in a title for an actual curative treatment) it makes it seem like it’s being sold as already here not sure how to best change it – adding ‘once available’? Rewording it in some way? ‘When we have HBV cure…’

The end paragraph of the introduction on first read implied that a cure was available, which is of course not true – yet. I think this needs to be more explicit in some form there. Actually, I’m noticing this similar implication/feel in many places and I think a few “potential’s” and “once available”’s may need to be added in a few places.

Line 239 – it might be worth spelling out what you mean by this statement even just to highlight the complexity of the potential regimens (oral vs infusions vs ??)– your target audience will be much broader than the usual basic science/clinical groups, so it probably warrants a little expansion.

Line 328 – “who are able to access cure” to “who would have access to cure” – I think the conditional tense is best used throughout since we are talking about the potential and not actual.

Line 366 – gender vs sex – confirm which it is you mean, or perhaps it is both.

Line 404 – colonizing nations – confirm this is the correct term and use

Line 436 – awkward wording and what is meant by structures in this sentence?

Line 478-9 – awkward wording

Line 537-8 – sentence is almost leaving an idea hanging – may need to be teased out more or reworded.

Line 555 – need to state “once available” or some equivalent as it can again be misleading.

Minor spell checks:

Line 223 – strengthening

Line 329 – countries

Line 548 - sometimes

Author Response

Thank you to the reviewer for these comments. Our responses to specific points follow:
